# Silencing *GhJUB1L1* (*JUB1-like 1*) reduces cotton (*Gossypium hirsutum*) drought tolerance

Qian Chen[1,2,3], Chaoya Bao[3], Fan Xu[3], Caixia Ma[3], Li Huang[3], Qigao Guo[1,2]*, Ming Luo[3]*

1 Key Laboratory of Horticulture Science for Southern Mountains Regions of Ministry of Education, College of Horticulture and Landscape Architecture, Southwest University, Chongqing, China, 2 State Cultivation Base of Crop Stress Biology for Southern Mountainous Land, Academy of Agricultural Sciences, Southwest University, Chongqing, China, 3 Key Laboratory of Biotechnology and Crop Quality Improvement, Ministry of Agriculture/Biotechnology Research Center, Southwest University, Chongqing, China

These authors contributed equally to this work.
* luo0424@126.com (ML); qgguo@126.com (QG)

**Data Availability Statement:** All relevant data are within the manuscript and its Supporting information files.

**Funding:** This research was funded by the National Natural Science Foundation of China (31571722

## Abstract

Drought stress massively restricts plant growth and the yield of crops. Reducing the deleterious effects of drought is necessary for agricultural industry. The plant-specific NAC (NAM, ATAF1/2 and CUC2) transcription factors (TFs) are widely involved in the regulation of plant development and stress response. One of the NAC TF, JUNGBRUNNEN1 (JUB1), has been reported to involve in drought resistance in *Arabidopsis*. However, little is known of how the *JUB1* gene respond to drought stress in cotton. In the present study, we cloned *GhJUB1L1*, a homologous gene of *JUB1* in upland cotton. *GhJUB1L1* is preferentially expressed in stem and leaf and could be induced by drought stress. GhJUB1L1 protein localizes to the cell nucleus, and the transcription activation region of which is located in the C-terminal region. Silencing *GhJUB1L1* gene via VIGS () reduced cotton drought tolerance, and retarded secondary cell wall (SCW) development. Additionally, the expression of some drought stress-related genes and SCW synthesis-related genes were altered in the *GhJUB1L1* silencing plants. Collectively, our findings indicate that *GhJUB1L1* may act as a positive regulator in response to drought stress and SCW development in cotton. Our results enriched the roles of NAC TFs in cotton drought tolerance and laid a foundation for the cultivation of transgenic cotton with higher drought tolerance.

## Introduction

Plants often suffer from multifarious biotic and abiotic stresses with a sessile lifestyle. Drought is one of the most devastating environmental factors which largely restrict the growth and yield of crop plants all over the world. Cotton (*Gossypium hirsutum*) is an important fiber and oil crop worldwide. The yield of cotton is strongly limited by drought stress. Water is essential in every phase of cotton growth and development. Therefore, breeding excellent drought-resistant cotton is the dream pursued by many breeding researchers.

and 31971984). The funders had no role in study design, data collection and analysis, decision to publish, or preparation of the manuscript.

**Competing interests:** The authors have declared that no competing interests exist.

Plants have evolved a range of tolerance mechanisms at the molecular, biochemical, physiological, and developmental levels to reduce water loss [1–4]. Many stress-related transcription factors (TFs) and their target genes formed a complex regulatory network for plant stress response, which had been extensively studied and been well reviewed previously [5–13]. In the drought response experiment, the recovery rate of drought resistance of the *ataf1-1* and *ataf1-2* mutants was 7 times higher than that of the wild type. At the same time, real-time quantitative PCR analysis found that the expression of *COR47*, *ERD10*, *KIN1*, *RD22* and *RD29A* related to drought stress response was enhanced [8]. It was found that the overexpression of *DgNAC1* significantly improved the salt tolerance of tobacco [7].

NAC TF is one of the largest families of plant unique transcription factorswhich were named as the *NAM* (*No Apical Meristem*) in *Petunia hybrida* [14] and the *ATAF1/2* and *CUC2* in *Arabidopsis thaliana* [15]. A large number of *NAC* genes were found in *Arabidopsis thaliana* [14], soybean (*Glycine max*) [16], wheat (*Triticum aestivum*) [17], alfalfa (*Medicago truncatula)* [18], cotton (*Gossypium spp.*), [19] and many other species, which play essential roles in response to abiotic and biotic stresses. JUNGBRUNNEN1 (JUB1, ANAC042) is a multifunctional member of the NAC TF family in *Arabidopsis*, which participates in the regulation of plant longevity and stress tolerance. The *JUB1*-overexpressing plants (*JUB1*ox) were tolerant to multiple stresses, while the *jub1-1* mutants showed hypersensitivity to these stresses. AtJUB1 directly regulates the expression of stress-responsive transcription factors such as *DREB2A*, and reduces the level of reactive oxygen species, which contributes to the enhancement of stress tolerance [20, 21]. It was recently reported that overexpression of *JUB1* also enhances drought tolerance in tomato [22] and banana [23]. AtJUB1 directly inhibits the expression of gibberellic acid (GA) and brassinosteroid (BR) biosynthetic genes, leading to the accumulation of DELLA protein, which inhibits growth and increases plant resistance to stresses [24, 25]. Reduced *SlJUB1* expression in tomato plants resulted in more sensitive to drought and exhibited higher levels of oxidative stress than that of control plants. In addition, ectopic expression of *AtJUB1* led to enhanced stress tolerance and reduced oxidative damage. Furthermore, *SlDREB1*, *SlDREB2* and *SlDELLA* are potential target genes of SlJUB1 under drought stress [22]. In conclusion, as a growth and stress response regulator, JUB1 provides great hope for genetic engineering to improve crop drought tolerance.

Here, we report that *GhJUB1L1* (*JUB1-like 1*) is a drought induced TF with a positive role in drought resistance in cotton. Compared with the control plants, the *GhJUB1L1* silenced cotton plants were more sensitive to drought stress. In addition, silencing *GhJUB1L1* reduces SCW synthesis in cotton. Our results provided new clues for further study of potential direct target genes for JUB1 TF.

## Materials and methods

### Plant materials

The cotton variety used in this experiment is Upland cotton J14 (*Gossypium hirsutum L.acc. Jimian14*) provided by Professor Ma Zhiying of Hebei Agricultural University. For gene expression analysis, the cotton plants were grown under natural field conditions in Chongqing. The cotton roots, stems, leaves and flowers (0 days post-anthesis, DPA) were cut and stored at -80˚C after quick-freezing with liquid nitrogen; For drought treatment, cotton seeds were planted in humus soil and cultured in plant light incubator at 25~28 ˚C. After the plants grew to 3 true leaves, they were kept away from water for 1 week. Normally watered plants were used as controls. The cotton aboveground parts of treatment group and control group were cut and stored at -80˚C after quick-freezing with liquid nitrogen. For VIGS experiment, cotton seeds were planted in humus soil and cultured in plant light incubator at 25~28 ˚C. The

cotyledons were infiltrated after they were fully expanded. The tobacco variety used for instantaneous expression is *Nicotiana benthamiana* and is preserved in our laboratory.

## Cloning of *GhJUB1L1*

The gene ID and the sequence of *GhJUB1L1* was downloaded from the Upland cotton genome database, CottonFGD (https://cottonfgd.org/). Primers were designed according to the downloaded sequence and the full length of the target gene was amplified using the cotton leaf cDNA as the template. The amplified primers were listed in S1 Table. Sequence alignment and phylogenetic analysis was performed using DNAMAN and MEGA7.0 software [26], respectively. The accession numbers are: GhJUB1-like (Gh_D06G2096), GhJUB1L1 (Gh_D06G2096.1), DzJUB1 (XP_022724113), HsJUB1 (KAE8725843), AtJUB1 (NP_181828), OsJUB1 (XP_015628846.1), and ZmJUB1 (XP_008644461.1). The evolutionary history was inferred using the Neighbor-Joining method [27]. The optimal tree with the sum of branch length = 286.31250000 is shown. The percentage of replicate trees in which the associated taxa clustered together in the bootstrap test (1000 replicates) are shown next to the branches [28]. The tree is drawn to scale, with branch lengths in the same units as those of the evolutionary distances used to infer the phylogenetic tree. The evolutionary distances were computed using the number of differences method [29] and are in the units of the number of amino acid differences per sequence.

## Real-time qRT-PCR analysis

Total RNA was extracted using the RNAprep pure Plant Kit (TIANGEN, Beijing, China). First-strand cDNA was synthesized using the PrimeScript RT reagent Kit with gDNA Eraser (TAKARA, Kyoto, Japan). qRT-PCR analysis was performed using Novostar-SYBR Supermix (Novoprotein, Shanghai, China): 94 °C for 2 min followed by 40 cycles of 94 °Cfor 30 s, 56 °C for 30 s, and 72 °C for 1 min. Three biological repetitions were performed. Relative expression levels were calculated using the $2^{-\Delta\Delta Ct}$ method [30]. The specific primers of the selected genes and the internal control, HISTONE3 (GenBank accession No. AF024716), are listed in S2 Table.

## Subcellular localization

The full-length CDS (Coding sequence) of *GhJUB1L1* (without terminating codon) was ligated into the binary vector 35S::GFP (modified from pCAMBIA2300) (S1 Table). The resultant 35S::GhJUB1L1::eGFP plasmid was introduced into agrobacterium tumefaciens strain GV3101 and infiltrated into tobacco (*Nicotiana benthamiana*) leaves for transient assays. The GFP fluorescence was observed via a confocal laser scanning microscopy (Leica SP8).

## Analysis of transcriptional activity in yeast

The transcription activation assay in yeast was performed using Yeastmaker™ Yeast Transformation System 2 (Clontech). Different regions of GhJUB1L1 were cloned into pGBKT7 vector (S1 Table). These plasmids and the control pGBKT7 were introduced into yeast strain Y2HGold, and the transformed strains were cultured on either SD (Synthetic Dropout Media)/-Trp, TDO (Triple Dropout Supplements, SD/-Trp-His-Ade) and TDO/x-α-gal (SD/-Trp-His-Ade/x-α-gal) incubated at 30°C for 3 days to detect the response of GAL4 reporter.

## Virus-induced gene silencing

To silence *GhJUB1L1* through the VIGS (Virus-induced gene silencing) system mediated by tobacco rattle virus (TRV), a 228 bp sequences (S1 Table) were selected from the CDS of

*GhJUB1L1* and ligated into the pTRV2 vector. The agrobacterium (GV3101) containing pTRV1+TRV:GhJUB1L1 were co-infiltrated into the cotton cotyledon with the pTRV1 +TRV:00 group as the negative control. Silencing the chlorophyll biosynthesis gene GhCLA1 affected the synthesis of chlorophyll in cotton leaves, which resulted in the production of albino seedlings. Therefore, pTRV1+TRV:GhCLA1 group was used as the positive control in this study to test the effectiveness of the VIGS system. After the positive control leaves showed albino phenotypes (about 1 week after receiving injection), the expression level of *GhJUB1L1* in the silenced lines was detected by real-time qRT-PCR (2 weeks after receiving injection), and the down-regulated plants were selected for subsequent observation and research. For drought treatment, the *GhJUB1L1* silenced plants and controls were kept away from water for 10 days, normally watered plants were used as controls, then phenotypes were recorded. Then the materials were rewatered, phenotypes were recorded after 2 days.

## Determination of ion leakage

For ion leakage measurements, the second leaves in the top were immersed in 10 ml deionized water with a hole punch and soaked at room temperature for 12 h. Electrical conductivity (R1) was measured at 25 ˚C, using a conduct meter (Shanghai INESA & Scientific Instrument CO. LTD). The samples were boiled for 15 min, cooled down to 25˚C, and conductivity (R2) was measured again. Ion leakage was calculated through the expression RI/R2×100. Three independent experiments were performed.

## Determination of Relative Water Content (RWC)

Plant material leaves of the same position) was weighed (fresh weight, FW), then put in a Petri dish containing water and kept at room temperature for 24 h. Then, leaves were weighed again (turgid weight, TW). Then put the leaves into the drying box to dry and weigh the dry weight (DW). Relative water content (RWC) was calculated using the following formula: RWC [%] = (FW–DW)/(TW–DW)×100. Three independent experiments were performed.

## Staining of lignin and cellulose

Cut the the second internode stem (3 weeks after receiving injection) of the plant material. Place the material on a clean slide and slice it continuously to form many thin section, placing them in clear water for later use. For Phlorotriphenol/HCl staining method, put a drop of 5% phloroglucinol solution (0.05 g phloroglucinol dissolved in 1 mL 95% alcohol, keep in dark place) in the middle of a clean slide, pick a section into the dye solution, then put a drop of concentrated hydrochloric acid. Once the material turns red, cover the glass quickly and observe under a normal microscope (BX41TF). The degree of lignification of the material was accorded to the intensity of pigmentation. The volatilization of concentrated hydrochloric acid and ethanol leads to the precipitation of resorcinol, it is browning of the material. Therefore, the material can be temporarily placed in 75% (V/V) ethanol for photographic observation. For Toluidine blue staining method, the material was dyed with 0.05% toluidine blue (0.05 g toluidine blue dissolved in 100 mL water) for 1 min. For the rest, please refer to the resorcinol/ HCl staining method [31, 32].

## Dual-luciferase (Dual-LUC) assays

Dual-luciferase (Dual-LUC) assays were performed according to the manual instruction of the Dual-Luciferase Reporter Assay System kit (Promega, USA). The promoter deletion fragments were fused with firefly luciferases (LUC) gene in the pGreen0800-LUC vector [33] (S1 Table).

Effector construct was generated by introducing the *GhJUB1L1* gene into the overexpression vector pCambia2300-35S-eGFP. Mixes of the agrobacteria strains harboring effector and reporter (effector: reporter, 2: 1, v/v) were introduced into 4-wk-old tobacco leaves by infiltration [34]. After 40~48 h, leaves were collected for the dual-luciferase assay.

## Statistical data analysis

Data are presented as means ± SD. Statistical analysis were performed by the one-tailed Student's t-test. *, **, and *** indicates significant differences at $p < 0.05$, $p < 0.01$, and $p < 0.001$, respectively.

## Results

### Characters of GhJUB1L1

Searching the public database of upland cotton genome, we found that there are 16 *JUB1-like* genes in cotton. In order to clarify the roles of 16 *JUB1-like* genes in cotton growth, their tissue expression was analyzed using the Cotton FGD website (https://cottonfgd.org/) Gene Profile transcriptome database. The results (Fig 1) showed that the expression levels of the 16 *JUB1-- like* genes were generally low in ovules and fibers at different stages, and were only predominantly expressed in leaves and stems. The expression levels of *Gh_D06G2096* and *Gh_A06G1947* were the highest in stem, the important tissue for water transport, suggesting that *Gh_D06G2096* and *Gh_A06G1947* may be related to drought resistance in cotton. Based on the above analysis, *Gh_D06G2096*, whose expression level in stem was higher than *Gh_A06G1947*, was selected for further study, and was cloned and renamed as *GhJUB1L1* (*JUB1-like 1*).

 In order to clone the *GhJUB1L1* (*Gh_D06G2096*) gene, the specific primers were designed to amplify the full ORF of this gene. The resulting sequence is 948 bp (S1 Fig) in length and encodes a protein with 315 amino acid residues, with a predicted molecular weight of 36.8 kD and an isoelectric point of 6.219 (S2 Fig).

 GhJUB1L1 contains the typical sequence of NAC transcription factor (TF) (S2 Fig). Compared with other JUB1 proteins from *Arabidopsis thaliana*, durian (*Durio zibethinus*), and hibiscus (*Hibiscus syriacus*), the N-terminal of GhJUB1L1 is highly conservative while the C-terminal is variable (S2 Fig). In order to clarify the phylogenetic relationship of GhJUB1L1, we selected homologous proteins from dicotyledonous plants, cotton (*Gossypium hirsutum*), durian (*Durio zibethinus*), hibiscus (*Hibiscus syriacus*), and Arabidopsis (*Arabidopsis thaliana*) and monocotyledonous plants, rice (*Oryza sativa*) and maize (*Zea mays*) for phylogenetic analysis. The results (S3 Fig) showed that GhJUB1L1 was closely related to JUB1 protein of cotton, durian, hibiscus and *Arabidopsis*, but was far related to rice and maize. The JUB1 proteins from monocotyledons and dicotyledons were divided into two groups, indicated that the gene evolved independently after the differentiation of monocotyledons and dicotyledons.

### Expression patterns of *GhJUB1L1*

To explore the functions of *GhJUB1L1* in cotton, we analyzed the expression levels of *GhJUB1L1* gene in different tissues and organs. The results indicated that the expression level of *GhJUB1L1* was higher in stem and leaf, while it was lower in flowers and roots under non-stress conditions. The expression of this gene in these tissues increased when plants were subjected to drought (Fig 2). Given that stem and leaf are the key tissues for water transport and release (transpiration) in cotton, our expression results implying that *GhJUB1L1* might be important for drought stress for cotton.

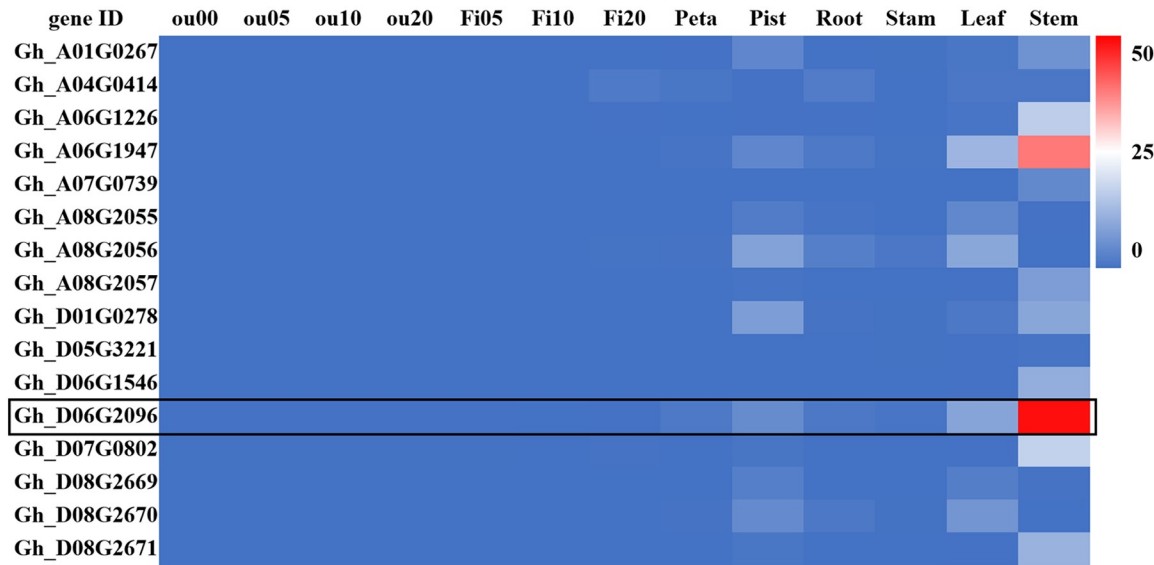

**Fig 1. Expression pattern heatmap of 16 *JUB1* genes in *Gossypium hirsutum*.** ou00, Ovule at 0 days post anthesis; ou05 to 0u20, Ovule at 5~20 days post anthesis; Fi05 to Fi25, Fibers at 5~20 days post anthesis. The unit of the color scale is FPKM.

## Subcellular localization of GhJUB1L1

To examine the subcellular localization of GhJUB1L1, we constructed the 35S::GhJUB1L1:: eGFP vector (S4 Fig), and introduced this vector in a transient expression system in *N. benthamiana* leaves by an agrobacterium-mediated method. The eGFP fluorescence signal was detected by a confocal laser scanning microscopy, and the results showed that the signal was found to be exclusively present in the nucleus, and was overlapped with the fluorescent signal of nuclear dye DAPI (Fig 3). These results indicate that GhJUB1L1 localizes in the nucleus.

## Transactivation activity assay of GhJUB1L1 in yeast

In order to identify whether GhJUB1L1 has transcriptional activity, the transcription activation assay in yeast was performed by using the Yeastmaker™ Yeast Transformation System 2 (Clontech). Different regions of *GhJUB1L1* CDS were cloned into pGBKT7 vector (Fig 4a).

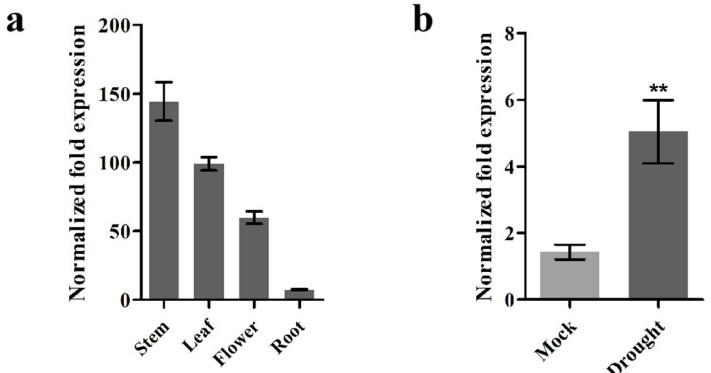

**Fig 2. Expression patterns of *GhJUB1L1*.** The expression patterns of *GhJUB1L1* in different tissues and organs (a) and in drought treatment (b). Error bars represent SD (standard deviation) of three independent replicates, ** indicates significant differences at $p < 0.01$.

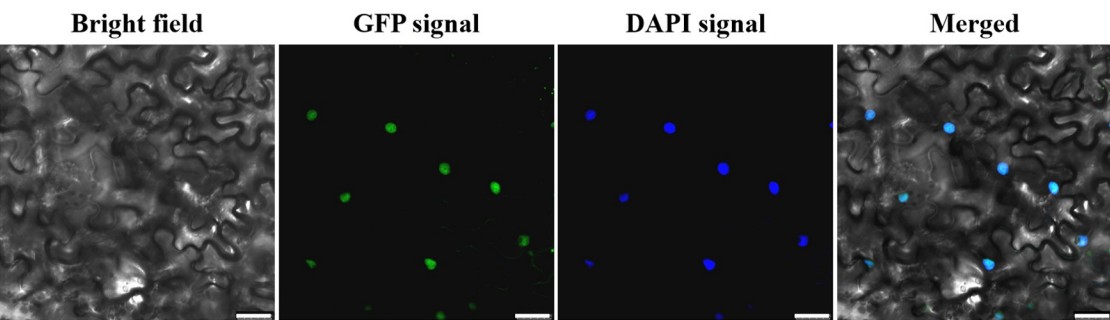

**Fig 3. The subcellular localization of GhJUB1L1.** DAPI, 4', 6-Diamidino-2-phenylindole dihydrochloride, fluorescent stain of nuclear; Merged, The merged images of Bright Field, eGFP and DAPI. Scale bar = 25 μm.

The results showed that yeast cells carryings BD -TAR (Transcriptional activation region) grew well in SD/-Trp, TDO and TDO/x-α-gal medium, and the x-α-gal chromogenic reaction presented blue color (Fig 4b), indicating that the TAR region of GhJUB1L1 had self-activation activity. Conversely, yeast cells carryings BD-full or BD-NAC domain could only grow on SD/-Trp medium. In summary, the GhJUB1L1 active region is located in the TAR region of its C-terminal. Its full-length transcriptional activity is inhibited, indicating that the N-terminal might play a regulatory role in its transcriptional activation activity.

## Silencing *GhJUB1L1* gene via VIGS reduces cotton drought tolerance

To elucidate the functions of *GhJUB1L1* in drought stress, we silenced the *GhJUB1L1* gene by the VIGS system. A 228 bp fragment was selected from the full length of *GhJUB1L1* gene and

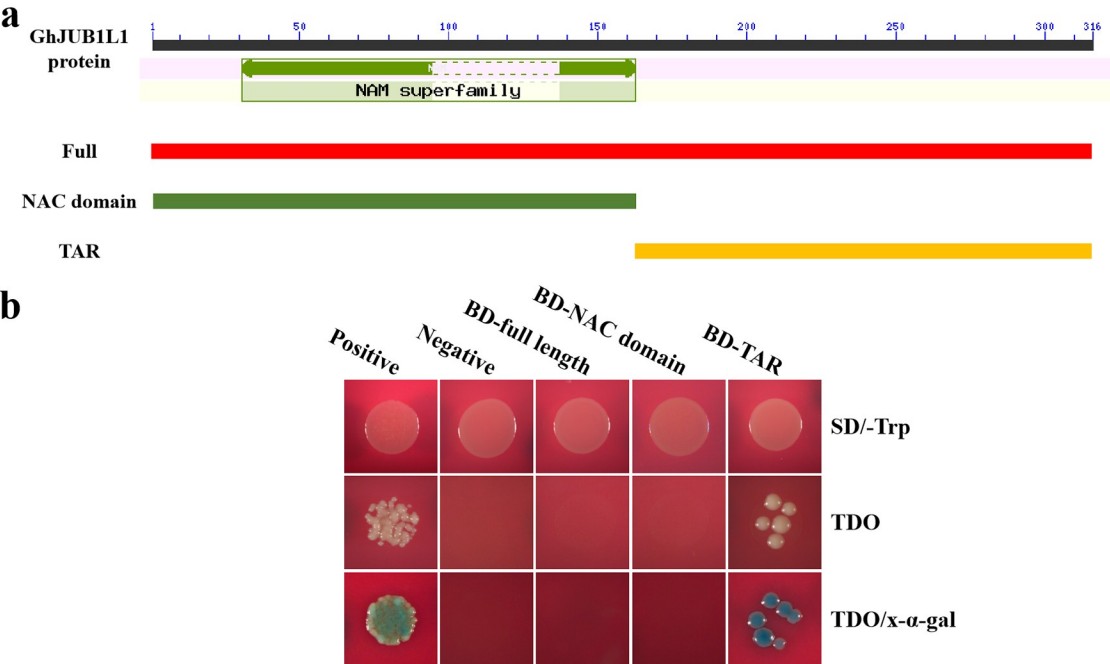

**Fig 4. Analysis of transcriptional activity of GhJUB1L1.** The structure of protein (a) and testing autoactivation of GhJUB1L1 (b). Full, Full length; TAR, Transcriptional activation region. Positive, positive control, represent pGBKT7-p53; Negative, negative control, represent pGBKT7; BD, pGBKT7 vector; TDO, Triple Dropout Supplements.

subjected to construct the TRV:GhJUB1L1 vector (S5 Fig). The cotton cotyledon was then impregnated with agrobacteria (GV3101) containing pTRV1 and TRV:GhJUB1L1, and pTRV:00 (pTRV1+pTRV2) was injected as a negative control. The *GhCLA1* gene affects the synthesis of chlorophyll in cotton leaves, silencing this gene leads to the production of albino seedlings. Therefore, TRV:GhCLA1 (pTRV1+TRV:GhCLA1) was used as a positive control. The results showed that after 1 week of injection, the leaves of TRV:GhCLA1 plants showed albino phenotypes, indicating that the VIGS system was operational (Fig 5a). We further detected the expression level of *GhJUB1L1* gene in *GhJUB1L1* silenced plants by qRT-PCR. Compared with the negative control, the transcript abundance of *GhJUB1L1* in *GhJUB1L1* silenced plants was significantly reduced, indicating that we obtained *GhJUB1L1* down-regulated cottons by this VIGS system (Fig 5b).

To further clarify the role of *GhJUB1L1* in drought tolerance of cotton, we silenced *GhJUB1L1* gene in cotton seedlings by VIGS. *GhJUB1L1* silenced plants and controls were kept away from water for 10 days (S6 Fig). As shown in Fig 7c, *GhJUB1L1* silenced plants appeared to be yellowing and more withered than controls which were still green. After re-watered, almost all control plants survived, as few silenced plants survived (Fig 5d and S7 Fig). Furthermore, RWC in leaves was determined after 10 days of withholding water. A significantly higher RWC was observed in control plants than in *GhJUB1L1* silenced plants (Fig 5e). Membrane stability under water deficit conditions was assessed by measuring ion leakage in both, control and treated plants. When water was withheld for 10 days, *GhJUB1L1* silenced plants showed a higher ion leakage than control plants (Fig 5f). Collectively, our results reveal that Silencing *GhJUB1L1* gene significantly reduced the drought tolerance of cotton plants.

## Silencing *GhJUB1L1* gene negatively regulates SCW development

The vascular bundle exists in stem, leaf (the vascular bundle in leaf is also called vein) and other organs, play crucial roles in the transport of water and minerals in plant tissues. To observe the xylem development of the *GhJUB1L1* silenced plants, we detected the change of lignin and cellulose by phloroglucinol-HCl and toluidine blue stain, respectively. Compared with the control group (Fig 6), the tracheary element was significantly smaller, and the pink of *GhJUB1L1* silenced plant xylem cells was weaker than that of control. Meanwhile, the intensity of toluidine blue stain staining of *GhJUB1L1* gene silencing plants tracheary element was lighter. The results were similar when we stained the re-watered cotton plants (S8 Fig). Based on above analysis, silencing *GhJUB1L1* gene can lead to decreased SCW deposition in plants tracheary element.

## Silencing *GhJUB1L1* gene suppressed the expression of drought-related and SCW synthesis-related genes

To further confirm the response of *GhJUB1L1* silenced plants to drought stress, we analyzed expression of several drought-related genes (such as *CBP1*, *P5CS*, *ABI1*, *SOS2*) in *GhJUB1L1* silenced plants and controls. As shown in Fig 7a–7d, the expression level of *ABI1* was increased, whereas the expression of *CBP1*, *P5CS*, and *SOS2* were reduced in the *GhJUB1L1* silenced plants. These results indicated that silencing *GhJUB1L1* gene can inhibit the expression of drought-related genes, suggesting that *GhJUB1L1* may be a positive regulator of cotton drought tolerance.

Silencing *GhJUB1L1* gene retarded tracheary element SCW development, we further detected the expression levels of genes involved in SCW formation. The results (Fig 7e–7k) showed that the expression levels of lignin biosynthesis genes *4CL1*, *CCoAOMT1*, and hemicellulose biosynthesis genes *IRX9*, *IRX14* were significantly down-regulated in the *GhJUB1L1*

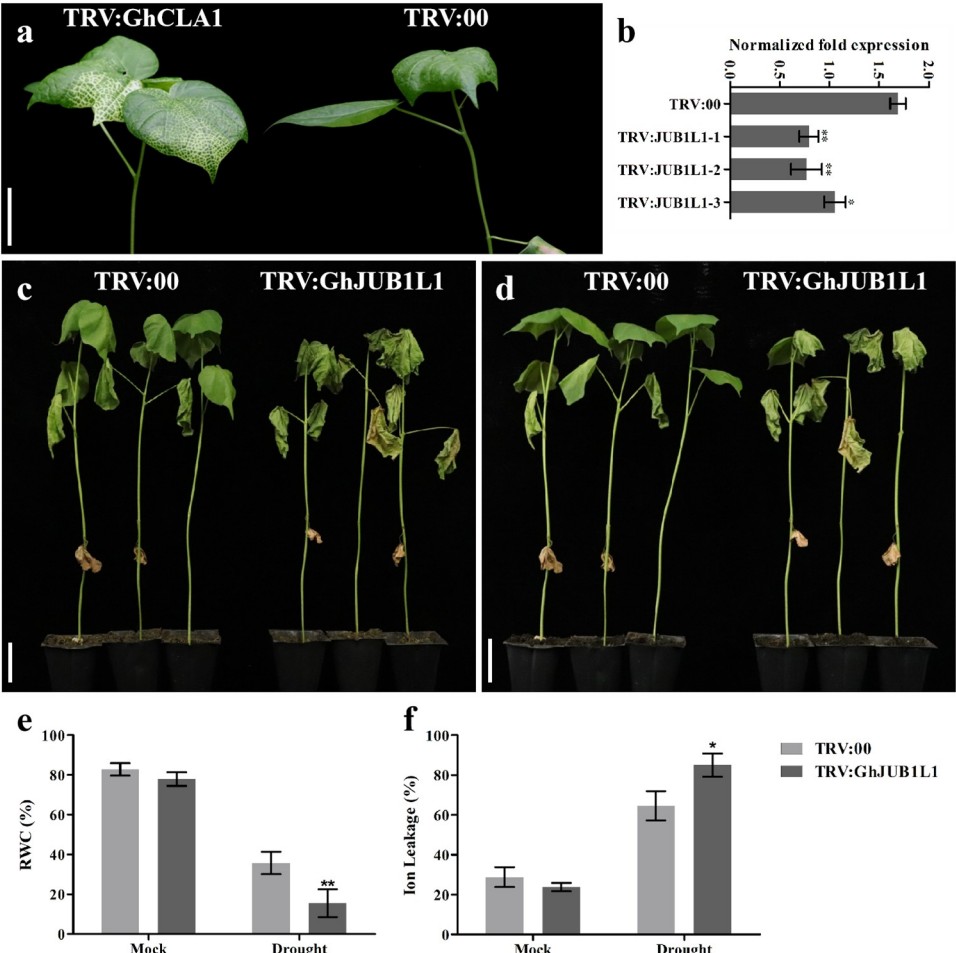

**Fig 5. Silencing *GhJUB1L1* gene reduces cotton tolerance to drought stress.** (a) Cotton phenotype after virus infection. (b) Relative transcription level of *GhJUB1L1* gene in TRV:GhJUB1L1 plants. (c-d) Phenotypic analysis of *GhJUB1L1* silenced plants and controls in drought stress and after re-watering. Scale bar = 5 cm. (e-f) Relative water content (RWC) ion leakage and of leaves (%). Error bars represent SD (standard deviation) of three independent replicates, * and ** indicates significant differences at p < 0.05, and p < 0.01, respectively.

silenced plants. Meanwhile, the expression levels of cellulose biosynthesis genes *CesA4*, *CesA7* and *CesA8* genes were also significantly down-regulated. These results indicated that silencing *GhJUB1L1* can inhibit the expression of genes involved in cellulose and lignin synthesis, suggesting that *GhJUB1L1* may be a positive regulator in SCW biosynthesis of cotton.

To verify whether GhJUB1L1 directly activates these genes expression *in vivo*, transient dual-luciferase assay was performed in tobacco epidermis cells. It was confirmed (Fig 8) that GhJUB1L1 can directly bind to the *GhABI1*, *GhSOS2*, *GhCCoAOMT1*, *GhCesA7*, and *GhIRX14* promoter, and then induce the expression of these five genes.

## Discussion

Plants should develop a complex regulatory circuit to respond to biotic and abiotic stresses, because the living environment is not always suitable. The regulatory circuit consists of transcriptional activators and repressors that regulate the expression of defense genes. In recent years, various NAC TFs in different plant species have been reported to be suitable targets for

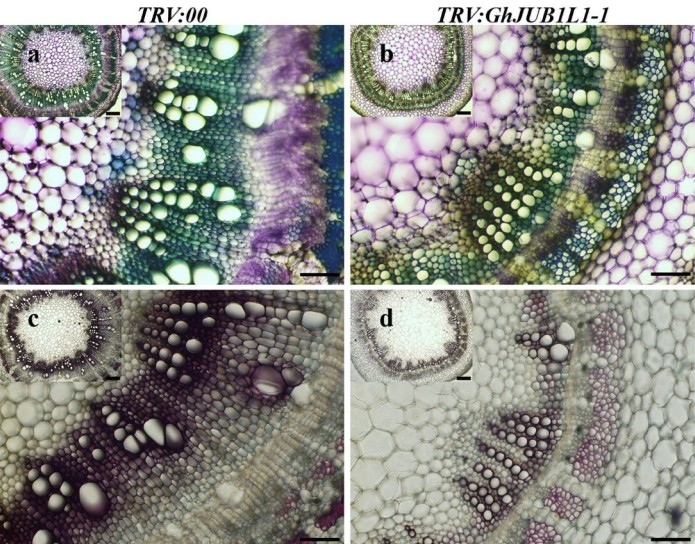

**Fig 6. Silencing *GhJUB1L1* gene inhibits SCW lignin in cotton.** Phloroglucinol/HCl (a-b) and toluidine blue staining analysis (c-d). Scale bar = 1 mm.

improving plant responses to dehydration/drought stress. For example, *anac016* mutants exhibit high drought tolerance, while those with *ANAC016* overexpression are sensitive to drought and exhibit accelerated aging [35]. In *Arabidopsis*, transgenic plants overexpressing *ANAC019*, *ANAC055*, or *ANAC072/RD26* show enhanced expression of stress response genes and improved tolerance to drought and salt stress [36]. The transgenic plants overexpressing *RD26* cDNA were hypersensitive to ABA, and inversely, the transgenic plants with *RD26* repressed were insensitive to ABA. The expressions of many ABA- and stress-induced genes including *RD20* and *GLY* genes were upregulated in plants overexpressing *RD26* and repressed in plants with *RD26* repressed. In *Arabidopsis* protoplasts, RD26 activated a promoter of the *GLY* gene that was upregulated in plants overexpressing *RD26* [37]. Microarray analysis of transgenic plants overexpressing either *ANAC019*, *ANAC055*, or *ANAC072* revealed that several stress-inducible genes were upregulated in the transgenic plants, and the plants showed significantly increased drought tolerance [36]. Microarray analysis of transgenic plants overexpressing *ZFHD1* revealed that several stress inducible genes were upregulated in the transgenic plants. Transgenic plants exhibited a smaller morphological phenotype and had a significant improvement of drought stress tolerance. Moreover, co-overexpression of the *ZFHD1* and *NAC* genes restored the morphological phenotype of the transgenic plants to a near wild-type state and enhanced expression of *ERD1* in transgenic *Arabidopsis* plants [38]. We isolated and identified *GhJUB1L1*, a homologue gene of *AtJUB1*. *GhJUB1L1* preferentially expressed in stem and leaf and could be induced by drought stress. These data give us a possible clue that *GhJUBIL1* may be involved in drought response. In this study, we found that cotton plants with silenced *GhJUBIL1* were more sensitive to drought stress than the control plants.

Given that TFs regulate the expression of stress-related target genes and participate in various plant stresses [39–41]. We analyzed the expression of several drought-related genes in *GhJUB1L1* silenced cottons. It has been reported that the expression of *CBF* genes in many plants increases significantly when subjected to abiotic stresses such as drought, high salinity, and exogenous hormones [42, 43]. Yamchi et al. [44] transferred *Arabidopsis P5CS* gene into tobacco. Under drought and salt stress, transgenic plants accumulated more proline and

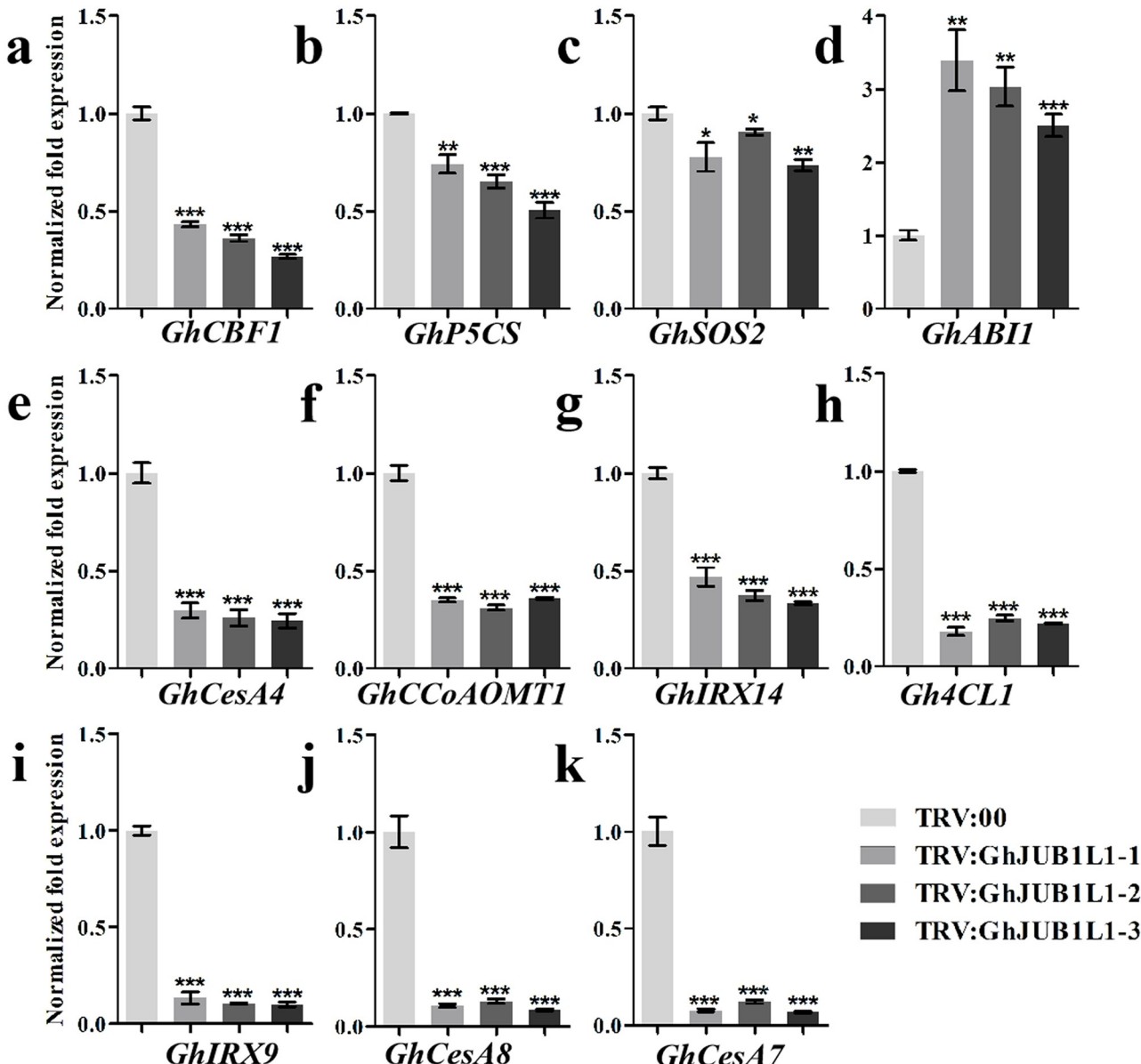

**Fig 7. Genes expression level in *GhJUB1L1* silenced plants.** The expression level of drought-related (a-d) and SCW-synthesis related genes (e-k). Error bars represent SD of three independent replicates. *, ** and *** indicates significant differences at p < 0.05, p < 0.01, and p < 0.001, respectively.

showed more stress tolerance than wide type control. Previous studies have shown that *SOS2* has the effect of improving drought resistance [45]. In this study, compared with the control, the expression levels of *CBF1*, *P5CS* and *SOS2* in *GhJUBIL1* silenced plants were significantly reduced. Type 2C protein phosphatase ABI1 (ABA insensitive 1) is a key negative regulator of ABA signal transduction and plays a negative role in stomatal closure induced by ABA [46–48]. The results of our analysis showed that the transcription level of *ABI1* in *GhJUB1L1* silenced plants was relatively higher than that in the control plants, suggesting that GhJUB1L1 may be a positive regulator of ABA signaling. These data suggest that GhJUBIL1 may be involved in drought response by regulating drought stress-related genes.

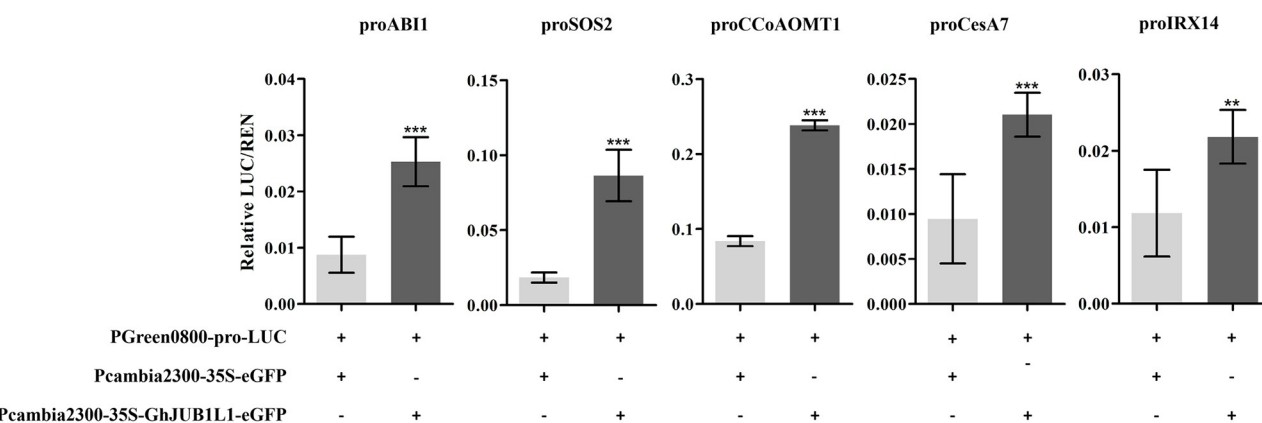

**Fig 8. Dual-LUC assay of GhJUB1L1 activated five genes promoter in *Nicotiana benthamiana* leaves.** Error bars represent the SD of three replicates. ** and *** indicates significant differences at p < 0.01 and p < 0.001, respectively.

SCW provide support for plants and forms a mechanical barrier against pathogen infection and abiotic stress. High expression of SCW biosynthesis related genes is necessary for plant tolerance to abiotic stress [49, 50]. Interestingly, the SCW development of *GhJUB1L1* silenced cottons was significantly reduced compared with the control, which resulted in decreased drought resistance of the plants. Therefore, we analyzed the expression of *CESA4*, *CESA7*, *CESA8*, *4CL*, *CCoAOMT1*, *IRX9*, and *IRX14*, and found that all these genes were significantly down-regulated in the *GhJUB1L1* silenced plants. In summary, our data suggest that *GhJUB1L1* may be involved in drought stress response and SCW development by regulating drought stress-related and SCW biosynthesis genes, which positively regulating plant response to drought stress.

## Supporting information

**S1 Fig. Cloning of *GhJUB1L1*.** M, Marker DL2000.
(TIF)

**S2 Fig. Multiple alignment comparison of GhJUB1L1 with other JUB1 proteins.** Horizontal lines, mazarine shading, and pink shading represent conserved NAC binding domains, conserved amino acid residues, and similar amino acid residues, respectively.
(TIF)

**S3 Fig. Evolutionary relationships analysis of GhJUB1L.** ▲ represents GhJUB1L1.
(TIF)

**S4 Fig. 35S::GhJUB1L1::eGFP vector construction.** CaMV35S, promoter.
(TIF)

**S5 Fig. TRV:GhJUB1L1 vector construction.** CaMV35S, promoter.
(TIF)

**S6 Fig. Field capacity (%).** Error bars represent SD (standard deviation) of three independent replicates.
(TIF)

**S7 Fig. Survival rates of the plants after re-watering.** Error bars represent SD of three independent replicates. *** represent p<0.001.
(TIF)

**S8 Fig. Silencing *GhJUB1L1* gene inhibits SCW lignin in cotton.** Phloroglucinol/HCl (a-b) and toluidine bluestaining (c-d) analysis for re-watered plants. Scale bar = 200 μm.
(TIF)

**S1 Table. Gene-specific primers used in isolation of *GhJUB1L1* genes and vector construction.**
(PDF)

**S2 Table. Gene-specific primers used in RT-PCR analysis.**
(PDF)

## Acknowledgments

We are grateful to professor Ma Zhiying (Hebei Agricultural University) for kindly providing the Jimian 14 seeds.

## Author Contributions

**Conceptualization:** Qian Chen, Qigao Guo, Ming Luo.

**Data curation:** Qian Chen.

**Investigation:** Qian Chen.

**Methodology:** Qian Chen, Chaoya Bao, Fan Xu.

**Software:** Qian Chen, Chaoya Bao, Caixia Ma, Li Huang.

**Validation:** Qigao Guo, Ming Luo.

**Writing – original draft:** Qian Chen, Chaoya Bao.

**Writing – review & editing:** Qian Chen, Chaoya Bao, Fan Xu, Qigao Guo, Ming Luo.

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
