## [Decision Letter · Decision Letter 0]

11 Aug 2021

PONE-D-21-21441

Silencing GhJUB1L1  ( J UB 1-like 1 ) reduces cotton  ( Gossypium hirsutum ) resistance to drought stress

PLOS ONE

Dear Dr. Luo,

Thank you for submitting your manuscript to PLOS ONE. After careful consideration, we feel that it has merit but does not fully meet PLOS ONE’s publication criteria as it currently stands. Therefore, we invite you to submit a revised version of the manuscript that addresses the points raised during the review process.

We look forward to receiving your revised manuscript.

Kind regards,

Ramegowda Venkategowda, PhD

Academic Editor

PLOS ONE

Journal Requirements:

Reviewers' comments:

Reviewer's Responses to Questions

**Comments to the Author**

1. Is the manuscript technically sound, and do the data support the conclusions?

Reviewer #1: Yes

Reviewer #2: Yes

2. Has the statistical analysis been performed appropriately and rigorously? 

Reviewer #1: Yes

Reviewer #2: Yes

3. Have the authors made all data underlying the findings in their manuscript fully available?

Reviewer #1: Yes

Reviewer #2: Yes

4. Is the manuscript presented in an intelligible fashion and written in standard English?

Reviewer #1: Yes

Reviewer #2: Yes

5. Review Comments to the Author

Reviewer #1: The authors have functionally validated the role of GhJUB1L1 in drought tolerance of cotton by transient down regulation using VIGS approach. GhJUB1L1 codes for a transcription factor belonging the NAC family and is shown to regulate stress responsive and secondary cell wall genes. Some of the comments to the authors are as follows:

Major comments:

In material and methods section: Please add reference for the protocols used for staining of cellulose and others. NO clarity of sample collection for qRT, staining etc. Mention after how many days of silencing symptoms (albino) the samples were collected for qRT, staining. After how many days of re-watering the phenotype was recorded. Drought stress imposition protocol is missing-after how many days of silencing the drought stress was imposed? Mention the % field capacity after 10days of withholding water.

Results section:

Need to provide physiological data eg. RWC or EC etc., to support that plant are experiencing stress and control plants are unstressed.

In Fig4 and other expression data-normalization against what?

Fig 4b- what is drought 1, 2 and 3?

What statistical parameter (ttest, ANOVA) was used is not mentioned.

It would be nice to have the staining of re-watered plants.

Gene name must be italicized throughout. It is better to use drought tolerance rather than drought resistance.

No discussion about why of the two genes expressed (heat map) in stem only one was chosen for study-criteria used?

In general, the language can be improved as it is not clear is most places. The sentences can be reframed to make it simple and clear.

Line 44-46-not clear

Line 56- remove and after wheat

Line 58- reframe as-which plays role in regulating longevity and stress tolerance

Line 70-reframe as-In addition, ectopic expression of AtJUB1 led to enhanced stress tolerance and reduced oxidative damage.

Line 73- growth and stress response interactor

Line 76- drought induced gene

Line 128- to silence

Line 130-remove and then,

Line 146-reframe- once the material turns red, cover the glass quickly and observe under a normal microscope.

Line 149- accorded to the intensity of pigmentation

Line 150-it is browning

Line 165-predominantly not predominance

Line 168-Based on above analysis, Gh…..was selected for further study which was cloned and renamed as GhJUB1L1

Line 209- the results indicated tthat expression level of GhJUB1L1 was higher in stem and leaf while it was lower in flowers and roots under non-stress conditions. The expression of this gene in these tissues increased when plants were subjected to drought.

Line 217- in/different

Line 225- in the nucleus

Line 276- not clear

Line 281, line 288-289- reframe

Line 353- a homolog instead of one homolog

Reviewer #2: Overall the design of the study is good and authors wrote conveyed the research findings very well. I have few suggestions for authors to improve the manuscript quality.

1. Please change all the figure legends (figure 1, 2, 3, 5, 6, 8 and 9). They were very hard to read during the review process. Also maintain same size and font for figure legends.

2. Other than qRT-PCR data of downstream targets. authors have not shown any direct evidence like yeast one hybrid assay or luciferase reporter assay with few downstream target gene promoters. It would add more strength to the research findings.

3. Please rename the GhCCOAOMT1 to GhCCoAOMT1 across the manuscript.

Thanks

6. PLOS authors have the option to publish the peer review history of their article (what does this mean?). If published, this will include your full peer review and any attached files.

Reviewer #1: **Yes: **Geetha Govind

Reviewer #2: **Yes: **Shailesh Karre

---

## [Author Response · Author response to Decision Letter 0]

28 Sep 2021

Journal Requirements:

Answer: Thank you for your careful comment. We have carefully checked our manuscript.

Answer: Thank you for your careful comment. We have carefully checked our manuscript.

Reviewer 1

Major comments:

In material and methods section: Please add reference for the protocols used for staining of cellulose and others. NO clarity of sample collection for qRT, staining etc. Mention after how many days of silencing symptoms (albino) the samples were collected for qRT, staining. After how many days of re-watering the phenotype was recorded. Drought stress imposition protocol is missing-after how many days of silencing the drought stress was imposed? Mention the % field capacity after 10days of withholding water.

Answer: Thank you for your careful comment. We have given detailed protocol in the Methods section. And the % field capacity after 10 days of withholding water was given in S6 Fig.

Results section:

1. Need to provide physiological data eg. RWC or EC etc., to support that plant are experiencing stress and control plants are unstressed.

Answer: Accepted and added.

2. In Fig4 and other expression data-normalization against what?

Answer: Thank you for your advice. The expression level of GhJUB1L1 gene in samples from root, mock, and TRV:2096-3 was set as 1 to figure 2a, 2b, and 5b, and then the expression levels of GhJUB1L1 gene in other samples was normalized to these for figure 2a, 2b, and 5b, respectively. The expression level of these genes in samples from TRV:00 was set as 1 to figure 7, and then the expression levels of these gene in other samples was normalized to these for figure 7, respectively.

3. Fig 4b- what is drought 1, 2 and 3?

Answer: Thank you for your careful comment. We have corrected.

4. What statistical parameter (ttest, ANOVA) was used is not mentioned.

Answer: Thank you for your good suggestion. We used t-test for statistical analysis and have added in the Methods section.

5. It would be nice to have the staining of re-watered plants.

Answer: Thank you for your valuable suggestions. We have added the staining images in S8 Fig.

6. Gene name must be italicized throughout. It is better to use drought tolerance rather than drought resistance.

Answer: Thank you for your advice. We have carefully checked our manuscript and used italics for genes, and replaced drought resistance to drought tolerance.

7. No discussion about why of the two genes expressed (heat map) in stem only one was chosen for study-criteria used?

Answer: Thank you for your valuable suggestions. We have explained it in the Results section.

In general, the language can be improved as it is not clear is most places. The sentences can be reframed to make it simple and clear.

Answer: Thank you for your valuable advices. We have carefully checked our manuscript.

8. Line 44-46-not clear

Answer: Thank you for your advice. We have corrected.

9. Line 56- remove and after wheat

Answer: Accepted and corrected.

10. Line 58- reframe as-which plays role in regulating longevity and stress tolerance

Answer: Accepted and corrected.

11. Line 70-reframe as-In addition, ectopic expression of AtJUB1 led to enhanced stress tolerance and reduced oxidative damage.

Answer: Accepted and corrected.

12. Line 73- growth and stress response interactor

Answer: Accepted and corrected.

13. Line 76- drought induced gene

Answer: Accepted and corrected.

14. Line 128- to silence

Answer: Accepted and corrected.

15. Line 130-remove and then,

Answer: Accepted and corrected.

16. Line 146-reframe- once the material turns red, cover the glass quickly and observe under a normal microscope.

Answer: Accepted and corrected.

17. Line 149- accorded to the intensity of pigmentation

Answer: Accepted and corrected.

18. Line 150-it is browning

Answer: Accepted and corrected.

19. Line 165-predominantly not predominance

Answer: Accepted and corrected.

20. Line 168-Based on above analysis, Gh…..was selected for further study which was cloned and renamed as GhJUB1L1

Answer: Accepted and corrected.

21. Line 209- the results indicated tthat expression level of GhJUB1L1 was higher in stem and leaf while it was lower in flowers and roots under non-stress conditions. The expression of this gene in these tissues increased when plants were subjected to drought.

Answer: Accepted and corrected.

22. Line 217- in/different

Answer: Accepted and corrected.

23. Line 225- in the nucleus

Answer: Accepted and corrected.

24. Line 276- not clear

Answer: Accepted and corrected.

25. Line 281, line 288-289- reframe

Answer: Accepted and corrected.

26. Line 353- a homolog instead of one homolog

Answer: Accepted and corrected.

Reviewer 2: 

Overall the design of the study is good and authors wrote conveyed the research findings very well. I have few suggestions for authors to improve the manuscript quality.

1. Please change all the figure legends (figure 1, 2, 3, 5, 6, 8 and 9). They were very hard to read during the review process. Also maintain same size and font for figure legends.

Answer: Thank you for your valuable suggestions. We have corrected.

2. Other than qRT-PCR data of downstream targets. authors have not shown any direct evidence like yeast one hybrid assay or luciferase reporter assay with few downstream target gene promoters. It would add more strength to the research findings.

Answer: Thank you for your valuable suggestions. We have added luciferase reporter assay with five downstream target gene promoters in Fig 8.

3. Please rename the GhCCOAOMT1 to GhCCoAOMT1 across the manuscript.

Answer: Accepted and corrected.

---

## [Decision Letter · Decision Letter 1]

19 Oct 2021

Silencing GhJUB1L1 (JUB1-like 1) reduces cotton (Gossypium hirsutum) drought tolerance

PONE-D-21-21441R1

Dear Dr. Luo,

We’re pleased to inform you that your manuscript has been judged scientifically suitable for publication and will be formally accepted for publication once it meets all outstanding technical requirements.

Kind regards,

Ramegowda Venkategowda, PhD

Academic Editor

PLOS ONE

Additional Editor Comments (optional):

Reviewers' comments:

Reviewer's Responses to Questions

**Comments to the Author**

1. If the authors have adequately addressed your comments raised in a previous round of review and you feel that this manuscript is now acceptable for publication, you may indicate that here to bypass the “Comments to the Author” section, enter your conflict of interest statement in the “Confidential to Editor” section, and submit your "Accept" recommendation.

Reviewer #1: All comments have been addressed

Reviewer #2: All comments have been addressed

2. Is the manuscript technically sound, and do the data support the conclusions?

Reviewer #1: Yes

Reviewer #2: Yes

3. Has the statistical analysis been performed appropriately and rigorously? 

Reviewer #1: Yes

Reviewer #2: Yes

4. Have the authors made all data underlying the findings in their manuscript fully available?

Reviewer #1: Yes

Reviewer #2: Yes

5. Is the manuscript presented in an intelligible fashion and written in standard English?

Reviewer #1: Yes

Reviewer #2: Yes

6. Review Comments to the Author

Reviewer #1: The expression data seems to be normalised with Histone 3, and the data for control (Unstressed) is mentioned in the graph. Therefore the 2-delta Ct is used, accordingly change in material and methods section.

Reviewer #2: (No Response)

7. PLOS authors have the option to publish the peer review history of their article (what does this mean?). If published, this will include your full peer review and any attached files.

Reviewer #1: **Yes: **Dr. Geetha Govind

Reviewer #2: **Yes: **Shailesh Karre

---

## [Editor Report · Acceptance letter]

28 Oct 2021

PONE-D-21-21441R1 

Silencing GhJUB1L1 (JUB1-like 1) reduces cotton (*Gossypium hirsutum*) drought tolerance 

Dear Dr. Luo:

I'm pleased to inform you that your manuscript has been deemed suitable for publication in PLOS ONE. Congratulations! Your manuscript is now with our production department. 

Kind regards, 

on behalf of

Dr. Ramegowda Venkategowda 

Academic Editor

PLOS ONE